# DYF-4 regulates patched-related/DAF-6-mediated sensory compartment formation in *C. elegans*

Hui Hong[1,2,3], Huicheng Chen[1,4,5], Yuxia Zhang[6,7], Zhimao Wu[1,4,5], Yingying Zhang[1,4,5], Yingyi Zhang[6], Zeng Hu[6], Jian V. Zhang[5], Kun Ling[6], Jinghua Hu[6], Qing Wei[5]*

1 CAS Key Laboratory of Insect Developmental and Evolutionary Biology, CAS Center for Excellence in Molecular Plant Sciences, Institute of Plant Physiology and Ecology, Chinese Academy of Sciences, Shanghai, China, 2 Department of Thoracic Surgery, Fudan University Shanghai Cancer Center, Shanghai, China, 3 Department of Oncology, Shanghai Medical College, Fudan University Shanghai Cancer Center, Shanghai, China, 4 University of Chinese Academy of Sciences, Beijing, China, 5 Center for Energy Metabolism and Reproduction, Institute of Biomedicine and Biotechnology, Shenzhen Institute of Advanced Technology, Chinese Academy of Sciences (CAS), Shenzhen, China, 6 Department of Biochemistry and Molecular Biology, Mayo Clinic, Rochester, Minnesota, United States of America, 7 Department of Cancer Biology, UT MD Anderson Cancer Center, Houston, Texas, United States of America

⊛ These authors contributed equally to this work.
* qing.wei@siat.ac.cn

**Data Availability Statement:** All relevant data are within the manuscript and its Supporting Information files.

## Abstract

Coordination of neurite extension with surrounding glia development is critical for neuronal function, but the underlying molecular mechanisms remain poorly understood. Through a genome-wide mutagenesis screen in *C. elegans*, we identified *dyf-4* and *daf-6* as two mutants sharing similar defects in dendrite extension. DAF-6 encodes a glia-specific patched-related membrane protein that plays vital roles in glial morphogenesis. We cloned *dyf-4* and found that DYF-4 encodes a glia-secreted protein. Further investigations revealed that DYF-4 interacts with DAF-6 and functions in a same pathway as DAF-6 to regulate sensory compartment formation. Furthermore, we demonstrated that reported glial suppressors of *daf-6* could also restore dendrite elongation and ciliogenesis in both *dyf-4* and *daf-6* mutants. Collectively, our data reveal that DYF-4 is a regulator for DAF-6 which promotes the proper formation of the glial channel and indirectly affects neurite extension and ciliogenesis.

## Author summary

In *C. elegans* sensory organ, the ciliated neuronal endings are wrapped in a luminal channel formed by glial cells, forming a specialized sensory compartment critical for sensory activity. Coordination of glial channel formation, dendritic tip anchoring and ciliogenesis are critical for the formation of a functional sensory compartment. DAF-6, a patched-related glial membrane protein, was reported to play an important role in glial channel morphogenesis, but the molecular function and regulatory mechanism of DAF-6 remain

**Funding:** This work was supported by National Natural Science Foundation Youth Project of China (Grant 31702019) to H.H., and National Natural Science Foundation of China (Grant 31671549) to Q.W. The funders had no role in study design, data collection and analysis, decision to publish, or preparation of the manuscript.

**Competing interests:** The authors have declared that no competing interests exist.

poorly understood. Here, we found that DYF-4, a glia-secreted protein, interacts and colocalizes with DAF-6, and functions in a same pathway as DAF-6 to regulate sensory compartment formation. We propose that DYF-4 is a novel regulator for DAF-6 to control sensory compartment formation.

## Introduction

The sensory organ uses its ending to receive environmental stimuli. In many cases, the ciliated neuronal receptive endings and surrounding glial cells form compartmentalized sensory endings, as observed in *C. elegans*, *Drosophila* and the olfactory epithelium of mammals [1–4]. The proper formation of the sensory compartment requires interplay between glial cells and the ciliary endings of sensory neurons. The integrity of the glial compartment and the development of sensory neurons impact each other in a reciprocal manner [5–9]. However, little is known about the underlying mechanism.

*C. elegans* uses its sensory organs (amphid and phasmid) to sense environmental cues. The amphid sensory organ of the head consists of 12 sensory neurons, whose ciliated endings are wrapped by the ends of a sheath glia cell and a socket glia cell [10]. The cell bodies of sensory neurons are located in the pharyngeal bulb, with the dendrites extending to the anterior end of the nose and terminating with cilia. Both the sheath and socket glial cells extend in parallel along the dendrites of sensory neurons and form discrete tubular channels surrounding amphid cilia. Specifically, the proximal portion of most cilia is surrounded by the membrane of the sheath cell, while the distal portion of most cilia extends through a pore formed by the membrane of the socket cell. The structure of the phasmid of the tail is similar to that of the amphid, except that it contains only 2 sensory neurons.

The proper formation of the sensory compartment in *C. elegans* includes the compartmentalization of glial endings, neuron dendrite tip anchoring and cilia biogenesis. DAF-6, a patched-related membrane protein, is a key regulator of the compartmentalization of the glial endings in *C. elegans* [11]. DAF-6 functions in the very early stage to restrict the expansion of the glial compartment [12], and its activity can be antagonized by several suppressors, including the Nemo-like kinase LIT-1, the actin regulator WSP-1, some retromer components (SNX-1, SNX-3, VPS-29) and the Ig/FNIII protein IGDB-2 [12–14]. It is thought that DAF-6 and its suppressors control glia compartment morphogenesis by regulating vesicle dynamics of glial cells in a cell-autonomous manner. In *C. elegans*, dendrite extension in the amphid and phasmid occurs via a special "retrograde extension" process [7,15,16]. Specifically, glial and neuronal cells are first assembled into a polarized multicellular rosette at the site where the sensory complex will be assembled, then the tips of dendrites stay anchored and the cell bodies grow backward to elongate the dendrites. The tectorin-related proteins DEX-1 and DYF-7 form an anchorage in the extracellular matrix and play essential roles in dendrite tip anchoring [7]. The transition zone (TZ) at the ciliary base is also reported to be involved in dendrite elongation in *C. elegans* [15]. Combined mutations of individual genes from the Meckel-Gruber Syndrome (MKS) module and Nephronophthisis (NPHP) module result in severe defects in dendrite elongation [15,17–20]. Notably, proper dendritic anchoring and cilia biogenesis are also required for glial compartment morphogenesis. It has been demonstrated that sheath cells cannot extend, and glial channels are disrupted in *dyf-7* mutants [7,21]. It was also reported that the glia morphology is altered in cilia mutants [8,22], and the glia compartment formation regulators DAF-6 and its suppressors are mislocalized in ciliogenesis mutants [11,12].

However, little is known about the identity and source of the extracellular signals that coordinate the glial channel formation and dendritic morphogenesis.

In a whole-genome forward genetic screen, we identified a novel gene, *dyf-4*, whose mutants recapitulate the phenotypes of *daf-6*, exhibiting truncated dendrites and defective ciliogenesis in both amphid and phasmid neurons. We cloned DYF-4 and found that it encodes a glia-secreted protein that interacts and colocalizes with DAF-6. Interestingly, the dendritic and ciliary defects of *dyf-4* could be restored by reported *daf-6* suppressors (*wsp-1*, *lit-1* and *igdb-2*). Our results reveal that DAF-6 is also required for dendrite elongation and ciliogenesis of sensory neurons, DYF-4 and DAF-6 function in the same pathway to regulate the proper formation of the sensory compartment.

## Results

### *dyf-4* encodes *C13C12.2* and is required for dendrite extension in *C. elegans*

In *C. elegans*, amphid cilia and phasmid cilia are wrapped by sheath and socket cells and extend to the external environment. Defects in environmental exposure or ciliogenesis result in dye-filling defects (*dyf*) [23]. In a forward genetic screen searching for *dyf* mutants, we isolated two mutants, *jhu431* and *jhu500*, sharing identical phenotypes, with extremely short phasmid dendrites and slightly truncated amphid dendrites (Figs 1A–1F and 2A–2F). The shorter dendrites suggest that *dyf* are mainly caused by defects in dendrite elongation in these two mutants. *jhu431* was mapped to *c13c12.2*, a gene of unknown function. C13C12.2 encodes a 383 amino acid protein, predicted to contain a 16 amino acid signal peptide at its N-terminus and a PLAC (protease and lacunin) domain at its C-terminus (Figs 1A and S1A and S1B). The PLAC domain is a six-cysteine region of approximately 40 amino acids that is usually present at the C-terminus of various extracellular proteins [24–26]. *jhu431* carries a missense mutation affecting one of the conserved cysteines (C366Y) in the PLAC domain (Fig 1A). A complementation assay indicated that *jhu431* failed to complement *dyf-4(m158)*, a mutant isolated 20 years ago that has not yet been cloned. *dyf-4 (m158)* shows the same phenotype as *jhu431*, including very short phasmid dendrites and slightly shorter amphid dendrites (Fig 1B–1F). Genomic sequencing showed that *dyf-4(m158)* possesses a nonsense mutation (Q166stop) in the middle region of C13C12.2. The elongation defects of the dendrites observed in *dyf-4 (m158)* and *jhu431* were fully rescued by the transgenic expression of the wild-type (WT) *dyf-4* gene under the control of its own promoter but were not rescued by the C13C12.3$^{C366Y}$ mutant form (Fig 1B–1F), indicating that C13C12.2 is responsible for these defects and that C366 is critical for its function. Therefore, we refer to C13C12.2 as DYF-4 hereafter.

### *daf-6* mimics the phenotype of *dyf-4*

Through a SNP mapping strategy and whole-genome sequencing, the *jhu500* allele was mapped to *daf-6* as a 1-bp deletion (c.2272delC) that leads to a frameshift and loss of the C-terminal region (Fig 2A). A null allele of *daf-6 (e1377)* showed the same phenotype as *jhu500* and failed to complement *jhu500*. The introduction of the WT *daf-6* cDNA fully rescued the dendrite extension defects of *jhu500*, revealing that *jhu500* is a new allele of *daf-6* (Fig 2B–2F).

DAF-6 is a glial protein similar to the hedgehog membrane receptor Patched. It has been proved that DAF-6 regulates glial compartment morphogenesis in *C. elegans* [11,12]. The dendrite elongation defect described here is a new phenotype of *daf-6* mutants, indicating that DAF-6 is also involved in normal dendrite extension of sensory neurons. Previous studies have shown that the glial compartment defect in *daf-6* mutants progressively becomes more severe throughout the larval development and becomes extremely disorganized in adult [11,12,27]. Interestingly, we observed that the dendrite length of *daf-6* and *dyf-4* was

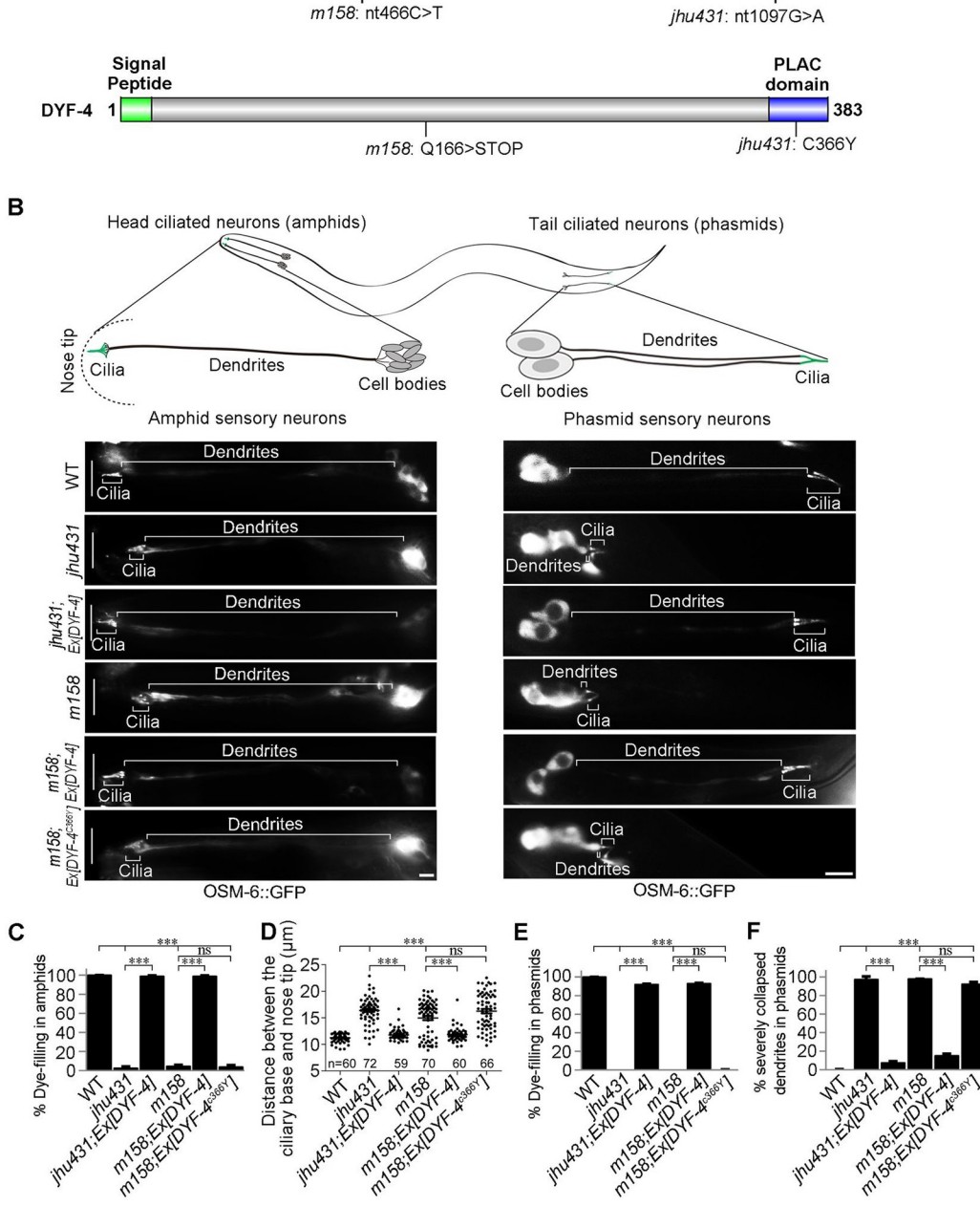

**Fig 1. *dyf-4* encodes *C13C12.2* and is required for dendrite extension in *C. elegans*.** (A) Schematic diagrams of the genomic and protein structures of *dyf-4* (*c13c12.2*). DYF-4 has a predicted signal peptide at its N-terminus and a PLAC domain at its C-terminus. *dyf-4* (*jhu431*) carries a point mutation (c.1097G>A) that leads to a missense mutation in one of the conserved cysteines (C366Y) in the PLAC domain. *dyf-4(m158)* possesses a point mutation (c.466C>T) that results in a nonsense mutation (Q166stop) in the middle region of the DYF-4 protein. As *dyf-4(jhu431)* and *dyf-4(m158)* exhibit the same mutant phenotypes, we used the *dyf-4(m158)* mutant in subsequent experiments if not otherwise specified. (B) Upper panels: schematic representation of *C. elegans* amphid and phasmid structures. Lower panels: Fluorescence micrographs of sensory neurons in the amphid and phasmid in adult WT, *dyf-4(jhu431)*, *jhu431*; *Ex [DYF-4]*, *dyf-4* (*m158*), *m158*; *Ex [DYF-4]* and *m158*; *Ex [DYF-4^C366Y^]* worms. The IFT-B component OSM-6::GFP was used to label cell bodies, dendrites and cilia. The white vertical line at left represents the nose tip position. Scale bars: 5 μm. (C) Statistics of the dye-filling percentages in the amphids of the indicated worm lines. (D) Statistics of the distance between basal bodies and the nose tip at the head of the indicated worm lines. (E) Statistics of the dye-filling percentages in the phasmids of the indicated worm lines. (F) Statistics of the ratio of phasmids with severely collapsed dendrites in the indicated worm lines.

Dendrites with cilia located near the cell bodies are defined as severely collapsed dendrites. Data are presented as the mean ± SEM from three independent experiments (n ≥ 80 per experiment). ***P < 0.001 (Fisher's Exact test for Fig 1C, 1E and 1F, Mann-Whitney test for Fig 1D).

comparable to that in WT at the L1 stage, but was significantly shorter than that of WT at the L2 stage (Fig 2G), indicating that the dendrite defect is also gradually severe during development and likely to be a secondary effect.

Dendrite elongation in *C. elegans* is mediated by "retrograde extension", with dendritic tips first being anchored to the extracellular matrix at the destination in the embryonic stage, after which the cell bodies travel backward during growth, trailing the elongating dendrites behind them [7]. Since the localization of neuron cell bodies is normal in both *daf-6* (*jhu500*) and *dyf-4* (*jhu431*), their short dendrites should be caused by the abnormal position of their dendritic tips. It is likely that the abnormal morphology of glial channels in *daf-6* [11] or *dyf-4* (see the data below) changes the anchoring position of the tips of dendrites, which indirectly leads to the shortening of dendrites.

## DYF-4 is a glial secretory protein required for DAF-6 localization

Due to the identical mutant phenotypes shared by *dyf-4* and *daf-6*, we speculate that DYF-4 may function in the same pathway as DAF-6 to regulate sensory compartment formation. To test our hypothesis, we first determined where *dyf-4* gene is expressed. To this end, a transcriptional reporter for *dyf-4* (*Pdyf-4*::GFP) was generated. Interestingly, as expected, we found that, similar to *daf-6*, *Pdyf-4*::GFP was expressed in sheath and socket glial cells in both the amphid and phasmid (Fig 3A and 3B). To further confirm that glial cells are the functional sites of *dyf-4*, we expressed *dyf-4* cDNA driven by the *daf-6* promoter or the neuron-specific *dyf-7* promoter. As expected, only the *daf-6* promoter-driven expression of DYF-4 rescued the *dyf-4* mutant phenotype (Fig 3C).

To observe the subcellular localization of DYF-4 protein, we created transgenic worms expressing GFP-tagged DYF-4 fusion proteins. Unfortunately, the DYF-4::GFP signal cannot be directly observed. To improve the signal of DYF-4::GFP, we first employed the indirect immunofluorescence method to enhance the signal by using the anti-GFP antibody. Interestingly, through this method, we observed that DYF-4::GFP was specifically enriched and partially co-localized with DAF-6 in the sensory compartment region in both amphids and phasmids (Fig 3D and 3E). Compared with DYF-4::GFP, the localization pattern of DYF-4$^{C366Y}$::GFP was changed and tended to accumulate in glia cells, suggesting that defects in *dyf-4 (jhu431)* mutants may arise from the mislocalization of DYF-4 (S2A Fig). Notably, without anti-GFP staining, DYF-4$^{\Delta(1-16)}$::GFP (the signal peptide deletion mutant of DYF-4) was directly observed to accumulate in vesicular structures in both amphid and phasmid (S2B Fig).

On the other hand, we replaced the GFP tag with an OPT-GFP tag which improves fluorescence and folding efficiency [28]. Interestingly, the DYF-4::OPT-GFP signal could be observed directly. Consistently, we observed that DYF-4::OPT-GFP signal was enriched in sensory compartment region in both amphids and phasmids (Fig 3F), showing puncta/granule patterns, suggesting that they are in vesicular structures (Fig 3G). Notably, DYF-4::OPT-GFP was expressed in the early developing glial cells of the embryo and tended to accumulate between the embryo and the eggshell, likely to be secreted (S2C Fig).

Next, we investigated the interdependence of the subcellular localization of the DYF-4 and DAF-6 proteins. In *daf-6* mutants, the localization of DYF-4::OPT-GFP was basically normal, only a fraction (~10%) of amphids had abnormal DYF-4 signals (Fig 4A). However, compared to WT in which DAF-6::GFP localized along the socket glial channel and showed weak signals

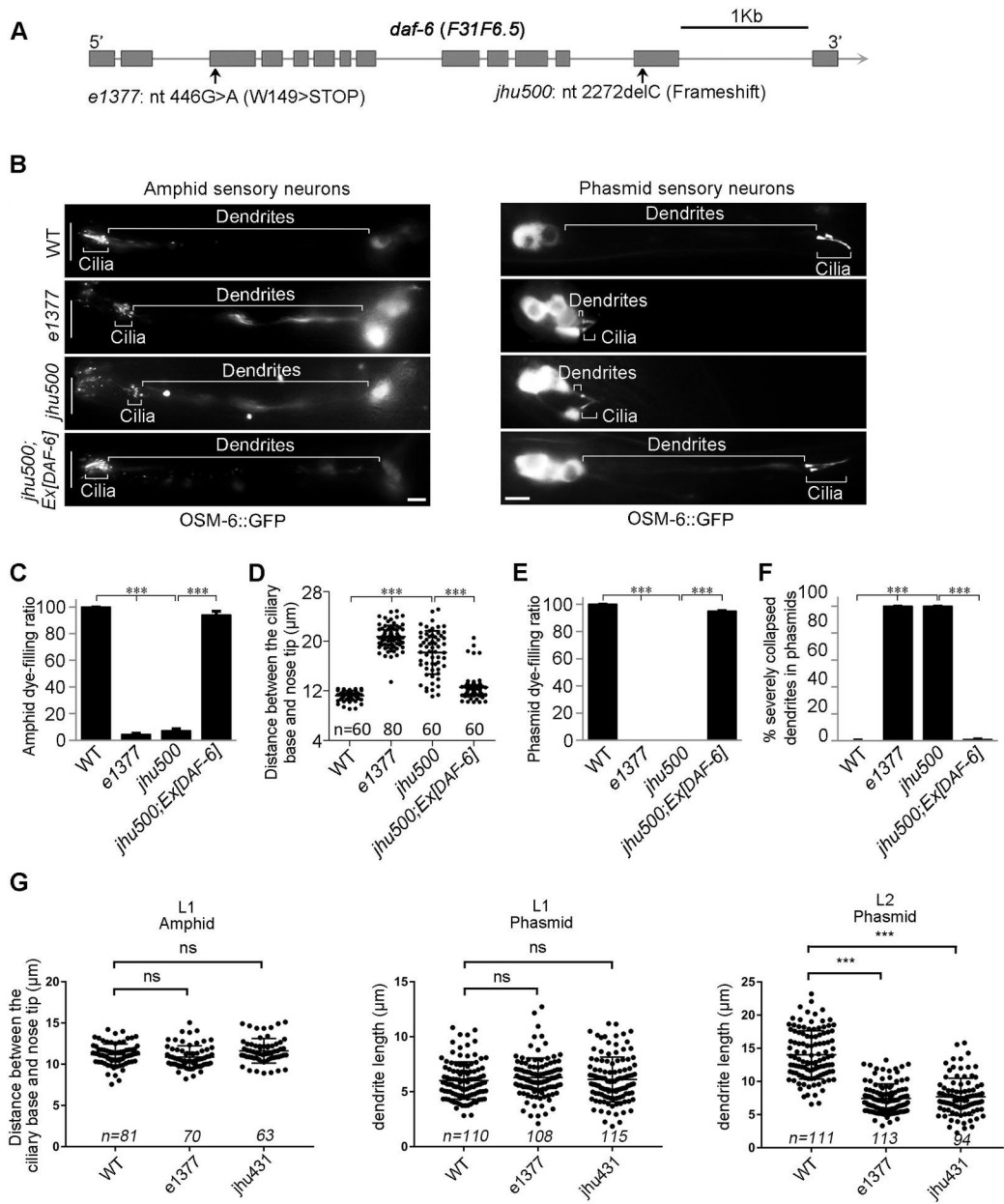

**Fig 2. The *daf-6* mutant mimics the phenotype of the *dyf-4* mutant.** (A) Schematic diagram of *daf-6* alleles. *daf-6(e1377)* was previously reported [11]. *daf-6(jhu500)* has a 1-bp deletion (c.2272delC) at its C-terminus, leading to a frameshift. (B) Fluorescence micrographs of sensory neurons in the amphids and phasmids of adult WT, *daf-6(e1377)*, *daf-6(jhu500)* and *jhu500*; *Ex [DAF-6]* worms expressing OSM-6::GFP. The white vertical line at left represents the nose tip position. Scale bars: 5 μm. (C) Statistics of the dye-filling percentages in the amphids of the indicated worm lines. (D) Statistics of the distance between basal bodies and the nose tip at the head in the indicated worm lines. (E) Statistics of the dye-filling percentages in the phasmids of the indicated worm lines. (F) Statistics of the phasmid dendrite defect ratio in the indicated worm lines. Dendrites with cilia located near the cell bodies are defined as severely collapsed dendrites. Data are presented as the mean ± SEM from three independent experiments (n ≥ 80 per experiment). (G) Analysis of the defects of dendrites at L1 and L2 stages in the indicated worm lines. ***P < 0.001 (Fisher's Exact test for Fig 2C, 2E and 2F, Mann-Whitney test for Fig 2D and 2G).

in the sheath lumen, DYF-6::GFP accumulated strongly in both cell bodies and sensory compartment region of glial cells in *dyf-4* mutants (Figs 4B and S2D), indicating that DYF-4 is required for the proper localization of DAF-6.

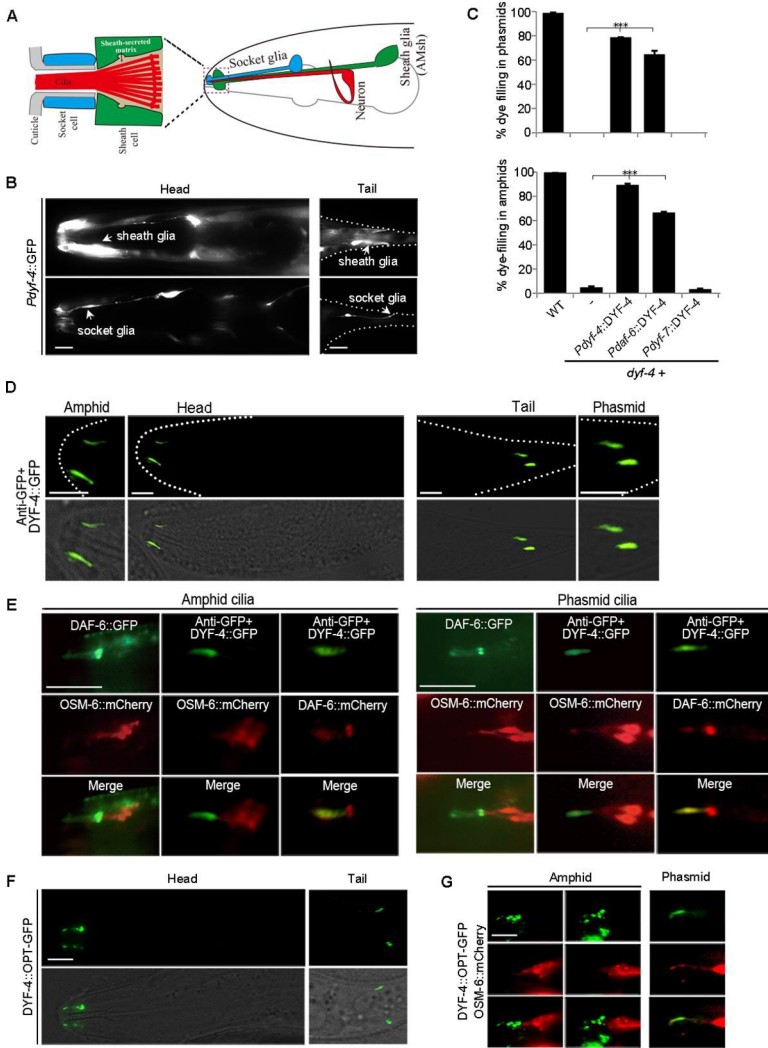

**Fig 3. DYF-4 is a glial secretory protein.** (A) Illustration of the *C. elegans* amphid. Right: Each amphid consists of 12 sensory neurons, a socket glial cell and a sheath glial cell. Left: Detail of the sensory compartment in the amphid. (B) Fluorescence micrographs of the *dyf-4* expression pattern in the head and tail region in worms expressing *Pdyf-4*::GFP. Sheath glial cells and socket glial cells are indicated by arrowheads. Scale bar: 5 μm. (C) The percentages of dye filling in amphid and phasmid in WT and *dyf-4(m158)* mutants expressing the WT *dyf-4* gene driven by different promoters. Data are presented as the mean ± SEM from three independent experiments (n ≥ 80 per experiment). ***P < 0.001 (Fisher's Exact test). (D) Representative images of DYF-4::GFP localization in the amphids and phasmids of young adult of WT. DYF-4::GFP localizes to the sensory compartment region. n = 30. Scale bar: 5 μm. (E) Colocalization of DYF-4 and DAF-6 in the amphid and phasmid of young adult of WT. n = 10. (F) The localization pattern of DYF-4:: OPT-GFP in the amphid and phasmid of young adult of WT. Scale bar: 5 μm. (G) Colocalization of DYF-4::OPT-GFP and OSM-6::mCherry in the amphid and phasmid of young adult of WT. DYF-4::OPT-GFP showed punctate localization and tended to accumulate at the middle of sensory compartment in amphids. Scale bar: 5 μm.

## DYF-4 is required for sensory compartment formation

DAF-6 is critical for the formation of amphid glia channels. Due to the abnormal glial channel, *daf-6* mutants often have curved cilia and vacuole accumulation in amphids [11]. Similarly, curved cilia and vacuolar structures were often observed in amphids of *dyf-4* mutants (Figs 4C and S3A). To visualize the sheath glia morphology, we labeled the amphid sheath glia with GFP driven by the T02B11.3 promoter and the ASER neuron with mCherry driven by the gcy-5 promoter. The ASER cilium extended through sheath glia in WT, but it was bent and fails to

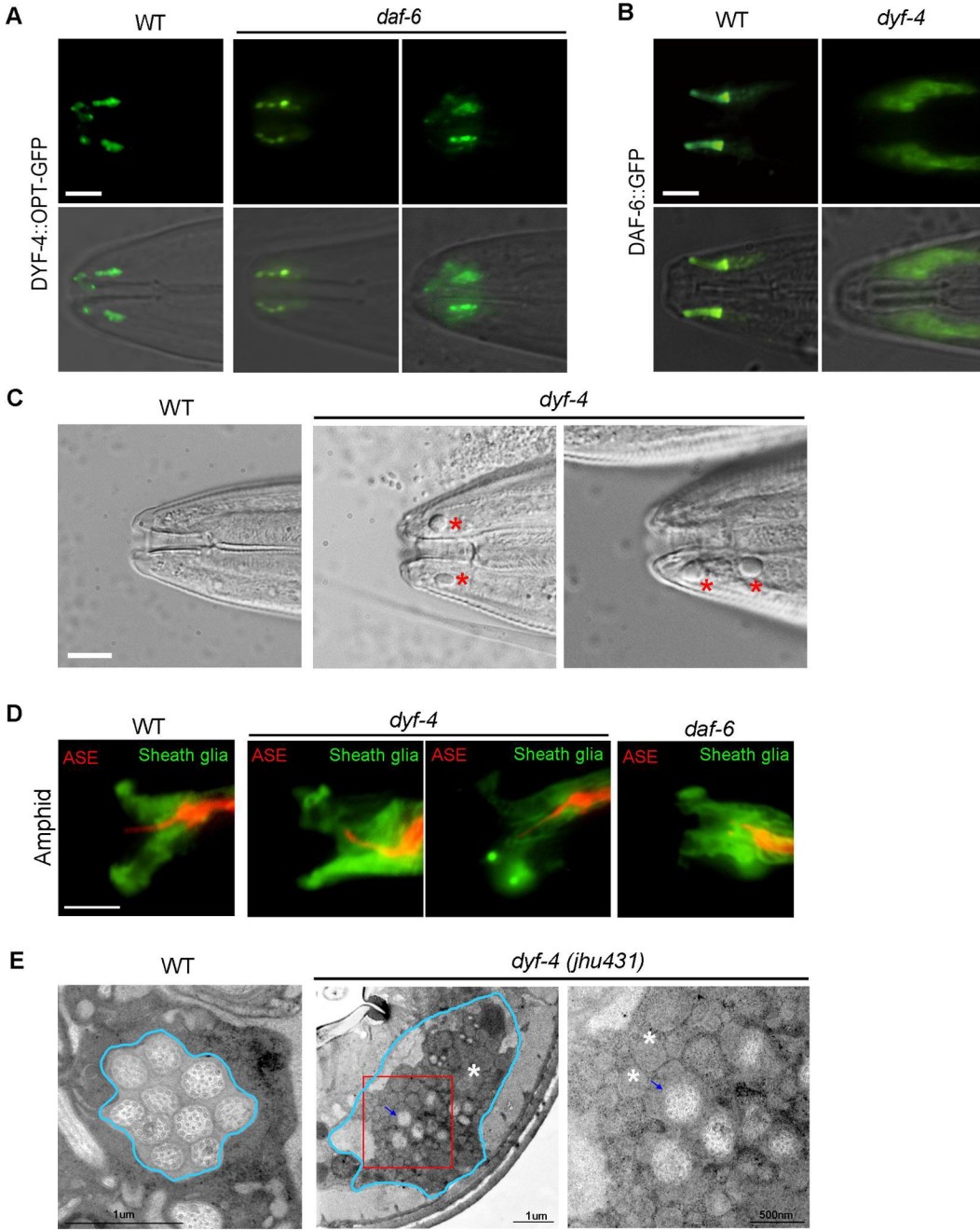

**Fig 4. DYF-4 is required for DAF-6 localization and glia channel formation.** (A) Subcellular localization of DYF-4::OPT-GFP in young adults of WT and *daf-6* worms. The localization of DYF-4 was basically normal in *daf-6* mutants. n>100 worms. (B) Subcellular localization of DAF-6::GFP in young adults of WT and *dyf-4* worms. DYF-4 is required for the glial channel localization of DAF-6. n>100 worms. Scale bars: 5 μm. (C) DIC images of heads of WT and *dyf-4* mutants. Vacuolar structures were observed in *dyf-4* mutants. Scale bars: 5 μm. (D) The amphid channel expressing *T02B11.3pro*::GFP (Green, sheath glia) and *gcy-5pro*::mCherry (Red, ASE neuron) in indicated genetic background. The ASE cilium extended through sheath cell lumen in WT, while it often bended and failed to extend through sheath cell lumen in both *dyf-4* and *daf-6* mutants. (E) TEM images of cross-sections of amphid channel in WT and *dyf-4* mutants. Blue arrow: cilia; White asterisk: vesicular accumulations. Magnified red square region in *dyf-4* mutants is shown on the right.

extend through sheath glia in *dyf-4* mutants (Fig 4D), similar results have been observed in *daf-6* mutants [12]. Dendrite defects are much more severe in phasmids of *dyf-4* mutants. we observed that sheath glia cells of *dyf-4* mutants fail to extend as they did in WT (S3B Fig). GFP::WSP-1A and ARX-2::GFP localize along the distal portion of the glial compartment channels (formed by socket glia) in WT worms [12,29]. Compared to WT, the signal length of WSP-1 and ARX-2 was significantly shortened in *dyf-4* mutants (S3C–S3H Fig). To directly examine the glia channel defects, we performed TEM on the amphid channel of *dyf-4* adult mutants. As showed in Fig 4E, enlarged pore lumen and accumulation of granule vesicles were observed in *dyf-4* mutants. All these results indicate that the glia channel is abnormal in *dyf-4* mutants.

## DYF-4 acts in a same pathway as DAF-6

We then wondered whether DYF-4 physically interacts with DAF-6. DAF-6 is a protein with 12 transmembrane domains, and both its N- and C-termini are predicted to reside in the cytoplasm. On the extracellular side, there are two longer extracellular loops (ECLs), ECL1 and ECL4. By using GST pull-down assay, we demonstrated that DYF-4 could specifically interact directly with both ECL1 and ECL4 of DAF-6 (Fig 5A), but not with ECL1 and ECL4 of CHE-14 (another 12 transmembrane glial protein containing sterol-sensing domain (SSD)) (Fig 5B), supporting the existence of an intriguing mechanism in which a complex formed by DYF-4 and DAF-6 is vital for the proper function of glial cells.

If DYF-4 acts in a same pathway as DAF-6, then the phenotype of *daf-4; daf-6* double mutant should be similar as either single mutant. Indeed, we found that co-deletion of *dyf-4* and *daf-6* did not aggravate dendrite elongation defects compared with either single mutants (Fig 5C).

Taken together, all these results support our hypothesis that DYF-4 and DAF-6 act in the same pathway to regulate sensory compartment formation.

## *daf-6* suppressors effectively restore *dyf-4* mutant phenotypes

Previous studies demonstrated that the Nemo-like kinase LIT-1, the actin regulator WSP-1, some retromer components and the Ig/FNIII protein IGDB-2 antagonize DAF-6 to promote glial compartment formation. The deletion of these genes also partially restores dye filling defects in the amphids of *daf-6* mutants [12–14]. We reasoned that if DYF-4 and DAF-6 function together to regulate glial function, *daf-6* suppressors should rescue phenotypes of *dyf-4* single mutants and *dyf-4; daf-6* double mutants. As expected, *wsp-1*, *lit-1* or *igdb-2* mutants all restored the dye-filling defects of *dyf-4* single mutants and *dyf-4; daf-6* double mutants, as they did in *daf-6* mutants (Fig 6A).

To confirm that the restoration of dye filling resulted from the rescue of the shortened dendrites in *dyf-4* and *daf-6* mutants, we focused on phasmids in which dendrites were severely shortened. In *dyf-4* or *daf-6* mutants, all the phasmid neuron dendrites are collapsed into the cell bodies. In contrast, in *dyf-4; wsp-1a* and *daf-6; wsp-1a* double mutants, only 3.19% and 5.55% of phasmid dendrites, respectively, were severely collapsed (Fig 6B and 6C). Consistent with these findings, the deletion of *lit-1*(*or131*) or *igdb-2*(*vc21022*) partially inhibited the dendrite elongation defects of *dyf-4* or *daf-6* single mutants (Figs 5B and 6C).

## Genetic interaction between glia compartment formation regulators, DYF-7 and the transition zone (TZ) in dendrite elongation

In addition to *dyf-4* and *daf-6* mutants, the phenomenon of shortened amphid or phasmid dendrites was previously reported in mutants of tectorin-related genes *dex-1* and *dyf-7*, and

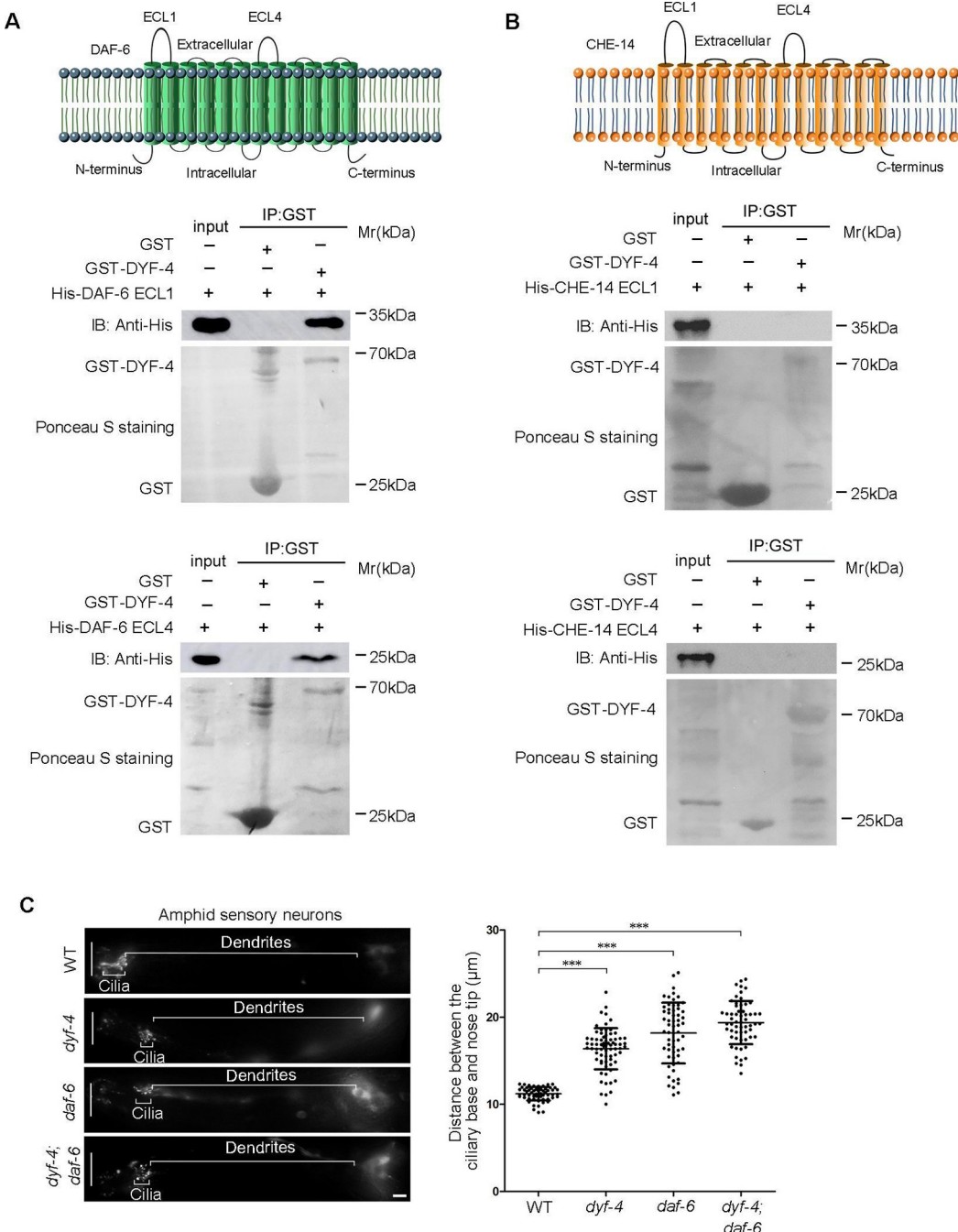

**Fig 5. DYF-4 interacts with DAF-6 and acts in a same pathway.** (A) DYF-4 interacted with DAF-6. Upper panel: Schematic representation of DAF-6 transmembrane protein. Lower panel: DYF-4 interacted with both DAF-6 ECL1 and DAF-6 ECL4 in the GST pull-down assay. (B) DYF-4 did not interact with ECLs of CHE-14 in the GST pull-down assay. (C) *dyf-4; daf-6* double mutants showed similar phenotypes with *dyf-4* or *daf-6* single mutants. The white vertical line at left represents the nose tip position. Statistics of the distance between basal bodies and the nose tip were showed on the right. All data are presented as the mean ± SEM (n ≥ 63 for each genotype). ***P < 0.001 (Mann-Whitney test).

mutants of certain combinations of TZ genes. DEX-1 and DYF-7 act in extracellular matrixes to anchor the dendritic tips or prevent the rupture of the connection between the sheath glia and the socket glia, deletion of either protein causes dendrite collapse [7,21]. Combined

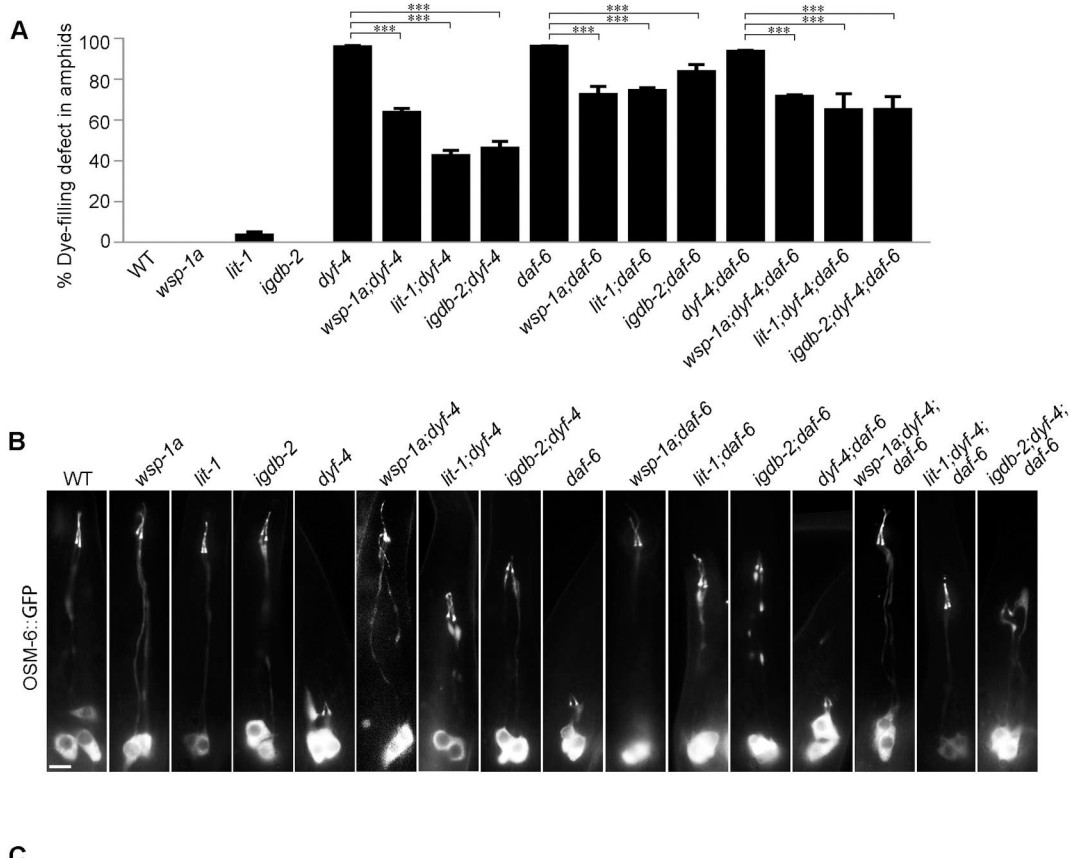

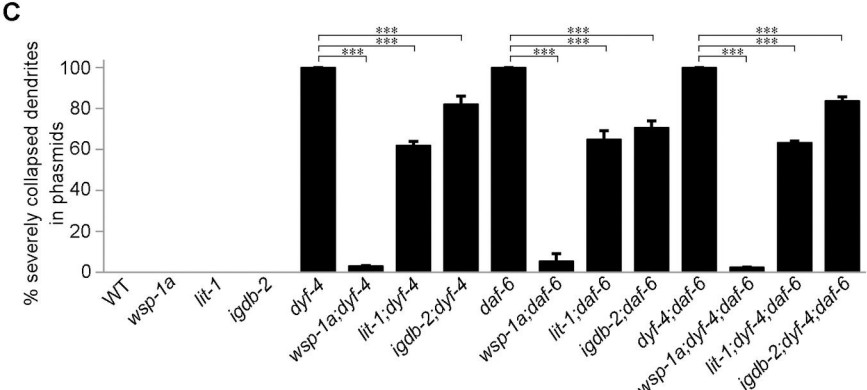

**Fig 6. Suppressors of *daf-6* inhibit defects in dendrite extension in both *dyf-4* and *daf-6* mutants.** (A) Statistics of the dye-filling defect percentages in the amphid neurons of the indicated worm lines. Suppressors of *daf-6*, *wsp-1a (gm324)*, *lit-1 (or131)* and *igdb-2* inhibited the defects in dendrite elongation in *dyf-4* and *daf-6* mutants. (B) Fluorescence micrographs of phasmid dendrites in the indicated genetic background. Scale bar: 5 μm. (C) The percentages of severely collapsed dendrites of the phasmid in the indicated worm lines. Dendrites with cilia located near the cell bodies are defined as severely collapsed dendrites. Data are presented as the mean ± SEM from three independent experiments (n ≥ 80 per experiment). ***P≤0.001 (Fisher's Exact test).

mutation of individual gene from CEP290, MKS module or NPHP module results in severe dendrite elongation defects in phasmid [15,17–20]. Although DYF-7/DEX-1 module and TZ proteins function at different developmental stages, and probably regulate the dendrite elongation through different mechanisms, the genetic interaction between them in phasmid dendrite length has been observed [15]. Therefore, we wondered whether there are genetic interactions

between glia compartment formation regulators and luminal ECM components or the ciliary TZ in dendrite extension. Since the phasmid dendrites in *dyf-4* or *daf-6* single mutants collapsed completely, they are not suitable for the synergetic interaction assay. Then we asked whether suppressors of *dyf-4* and *daf-6* could suppress the phenotype of *dyf-7(ns117)* or *nphp-1; mks-6* double mutants. To this end, we generated *wsp-1; dyf-7* double mutants and *wsp-1; nphp-1; mks-6* triple mutants. Interestingly, we did observe that *wsp-1* significantly suppressed the dendrite collapse in phasmid in either *dyf-7* or *nphp-1; mks-6* double mutants (S4A Fig), suggesting that there are genetic interactions (even probably indirectly) between glia compartment morphogenesis, dendrite anchoring complex and the TZ in dendrite extension.

## The ciliogenesis defects of *dyf-4* and *daf-6* mutants can be rescued by suppressors

In addition to the phenotype of short dendrites, ciliogenesis is also defective in *dyf-4* and *daf-6* mutants. The cilia are significantly shorter and usually lack distal segments in both types of mutants (Fig 7A). The length of cilia is approximately 7 μm on average in WT, while it is only approximately 3 μm in both mutants (Fig 7B). The intraflagellar transport (IFT) machinery is essential for cilia biogenesis [30]. The examination of various IFT components, including the IFT-B components OSM-6 (the homolog of IFT52) and OSM-5 (the homolog of IFT88), the IFT-A component CHE-11 (the homolog of IFT14) and the BBSome component BBS-7, showed that the ciliary fluorescence levels of all these IFT components were dramatically decreased, suggesting that the ciliary entry of IFT was compromised in *dyf-4* and *daf-6* mutants (S5A Fig). Notably, the observed ciliogenesis defects could be fully rescued by the transgenic expression of the wild-type *dyf-4* or *daf-6* genomic sequence (Fig 7A and 7B). How glial dysfunction indirectly impacts IFT behavior and ciliogenesis in sensory neurons remains an open question.

If the ciliary defects in *dyf-4* or *daf-6* mutants are non-cell autonomous consequences caused by dysfunctional glial cells, glial suppressors of *dyf-4* and *daf-6* might also rescue ciliogenesis defects. Consistent with this hypothesis, deletion of *wsp-1*, the suppressor of glial compartment formation of *daf-6* mutants, indeed restored cilia length in *dyf-4* or *daf-6* mutants (Fig 7C and 7D).

## Discussion

Previous studies demonstrated that the patched-related transmembrane protein DAF-6 functions in glia cells to negatively control the sensory compartment pore size [11]. Here, we identified a glial secretory protein DYF-4 as an apparent regulator of DAF-6. We showed that *dyf-4* mutants completely recapitulate the phenotype of *daf-6* mutants, and the suppressors of *daf-6* mutants can also rescue defects of *dyf-4* mutants. We demonstrated that DYF-4 interacts with DAF-6 and is required for the localization of DAF-6. Our results strongly indicate that DYF-4 functions in a same pathway as DAF-6 to regulate the sensory compartment formation (Fig 8).

It was originally proposed that DAF-6 is required for sensory pore lumen opening [11]. Later, it was discovered that DAF-6 may not be required for pore lumen opening, but restricts the pore lumen size [12]. EM studies of amphids in *daf-6* mutants indicated that the initial stages of sheath and socket channel development are unperturbed, but sheath lumen starts bloating from embryo about 420 min post-fertilization (before cilia have formed) and gradually becomes more and more serious as it grows up [12]. Starting from L2, the sheath and socket channels are no longer continuous, and dendrite endings become disorganized [12,27]. As dendrite tips are anchored in the extracellular matrix of pore lumen, disorganized pore lumen will certainly affect the position of dendrite tips. Therefore, it is plausible to speculate

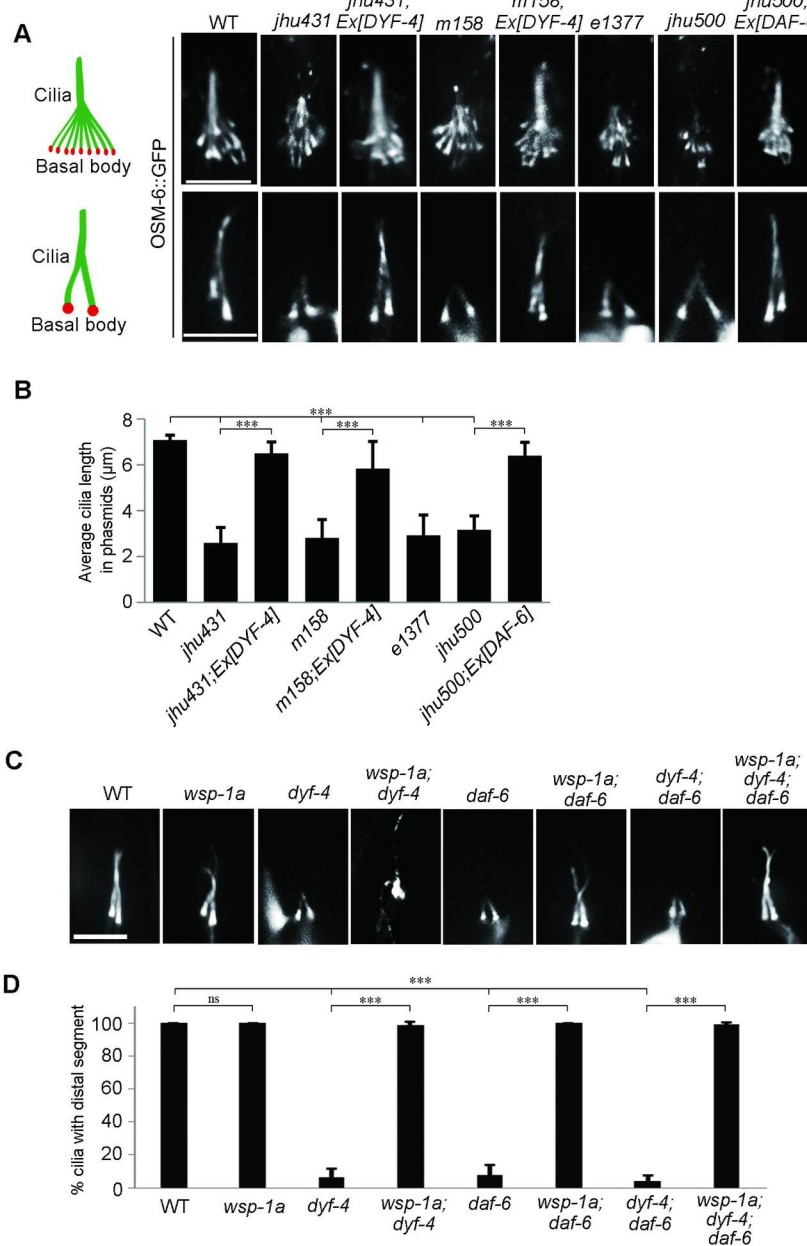

**Fig 7. Ciliogenesis is compromised in *dyf-4* and *daf-6* mutants and can be rescued by *wsp-1*.** (A) Fluorescence micrographs of amphid and phasmid cilia in the indicated worm lines expressing OSM-6::GFP. (B) Phasmid cilia length quantification in the indicated worm lines. (C) Fluorescence micrographs of phasmid cilia in the indicated genotypes. (D) Percentage of severely collapsed dendrites in the phasmid in the indicated worm lines. Scale bars: 5 μm. Data are presented as the mean ± SEM (n ≥ 60 for each genotype). ***P < 0.001 (Mann-Whitney test for Fig 6B, Fisher's Exact test for Fig 6D).

that dendrite defects in *daf-6* mutants are a secondary phenotype as the amphid channel structures become increasing disorganized. This hypothesis was supported by our observation that reported glial suppressors of *daf-6* could suppress the dendrite defects of *dyf-4* and *daf-6* mutants. Since all the phenotypes of *dyf-4* mutants observed so far are similar to *daf-6* mutants, the short dendrite phenotype in *dyf-4* mutants should also be a secondary phenotype. In

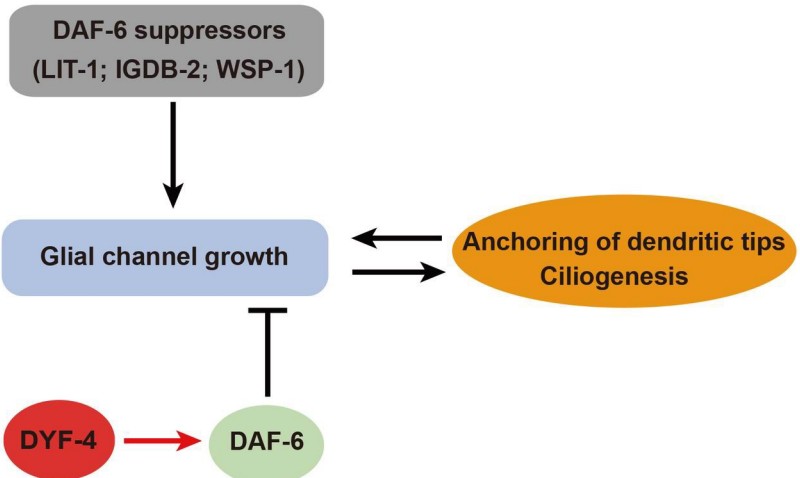

**Fig 8. A model for the role of DYF-4 and DAF-6 in sensory compartment morphogenesis.** The glial channel morphogenesis coordinates with the dendrite tip anchoring and ciliogenesis. DAF-6 restricts glial channel growth, while suppressors of *daf-6* promote it [12]. DYF-4 regulates the function of DAF-6, which in turn restricts the expansion of glia channel.

future, we need to use TEM and fluorescence visualization techniques to characterize the sheath and socket channel defects of *dyf-4* mutants at different stages of growth in more detail, and compared with *daf-6* mutants to test our hypothesis and understand the underlying mechanism.

The molecular mechanism of DAF-6 regulating glia morphology is still not clear yet. According to the literature, it is generally believed that DAF-6 acts as a receptor like Patched. It is thought that DAF-6 regulates the size of glia channel by controlling the amount of membrane deposited on the glia channel surface through regulating exocytosis and/or endocytosis [9,11,12]. If DAF-6 is indeed a receptor, DYF-4 may be a ligand/ regulator/chaperone of DAF-6, and it may cooperate with DAF-6 to control glia compartment formation by regulating exocytosis and/or endocytosis related vesicular dynamics. In addition to function as receptors, several SSD-proteins have been shown to act as transporters in intracellular vesicles. For example, NPC1 (Niemann-Pick disease, type C1), a protein containing SSD, cooperates with a secreted protein NPC2 to transport cholesterol in lysosomes [31]. Interestingly, DYF-4 is a secreted protein containing a signal peptide and is localized to punctate vesicular structures, and DAF-6 has previously been observed in intracellular vesicles within the cytoplasm in vulva and excretory canal cells [11]. Therefore, an attractive possibility is that, like the relationship between NPC1 and NPC2, DAF-6 and DYF-4 cooperatively function in intracellular vesicular compartments to regulate the sorting and recycling of certain cargos critical for vesicular dynamics and/or membrane formation. So far, there is no evidence to support that DAF-6 is a receptor yet. In addition, our result that the neuronal expressed DYF-4 could not rescue the defective phenotype of *dyf-4* mutants (Fig 3C) argues for a glial cell-autonomous requirement for DYF-4, which is inconsistent with the ligand-receptor relationship. In order to determine the molecular function of DAF-6 and DYF-4 in regulating the morphogenesis of glia channels, it will be necessary to conduct further detailed studies on the biochemical characteristics of DAF-6 and DYF-4, the characteristics and behaviors of intracellular vesicles of DYF-4 and their relationship with DAF-6 in future.

## Materials and methods

### *C. elegans* strains

The *C. elegans* strains used in this study are listed in S1 Table. The worms were cultured and maintained on NGM agar seeded with OP50 bacteria at 20˚C under standard conditions. The *dyf-4(jhu431)* and *daf-6(jhu500)* mutants were identified from an EMS screening for mutants showing dye filling defects. Standard genetic crossing was used to introduce reporter transgenes into worms of various genetic backgrounds and generate double or triple mutants. Genotyping was performed using PCR and DNA sequencing.

### Dye-filling assay

Worms were rinsed into a 1.5 ml EP tube with M9 buffer, followed by centrifugation in a desktop centrifuge at low speed. Then, the supernatant was removed, and the worms were washed twice with M9 buffer. After washing, the worms were incubated with 10 μg/ml DiI in M9 buffer at room temperature for 2 h and were then washed again with M9 buffer three times. Dye filling in the amphid was observed with a Nikon SMZ18 stereomicroscope, and dye filling in the phasmid was observed with a Nikon Eclipse Ti microscope with a Plan Apochromat 100× objective.

### Plasmids and transgene generation

To generate the *Pdyf-4::DYF-4::GFP* plasmid, a DNA fragment including the 1511 bp promoter and the whole genomic DNA sequence of *dyf-4* was cloned into the SmaI and KpnI restriction sites of the GFP vector pPD95.75. For the construction of *Pdyf-4::DYF-4$^{C366Y}$::GFP*, the guanine (G) nucleotide at site 1097 of the *dyf-4* CDS was replaced by adenosine (A). For the construction of *Pdyf-4::GFP*, 1511 bp of the promoter of *dyf-4* was amplified by PCR and ligated into the SmaI and KpnI restriction sites of pPD95.75. To generate *Pdaf-6::DAF-6::GFP*, 3096 bp of the promoter of *daf-6* was ligated into the BamHI and SmaI restriction sites of pPD95.75, and the CDS of *daf-6* was ligated into the SmaI and KpnI restriction sites of pPD95.75. To construct *Pdaf-6::DYF-4::GFP* and *Pdyf-7::DYF-4::GFP*, the *dyf-4* CDS was ligated into the SmaI and PstI restriction sites of pPD95.75, and 3096 bp of the promoter of *daf-6* or 1758 bp of the promoter of *dyf-7* was ligated into the PstI and SmaI restriction sites of pPD95.75. To obtain transgenic worms, plasmids were injected into wile-type animals at 30 ng/μl with the dominant roller marker *pRF4 [rol-6(su1006)]* at 30 ng/μl.

### Microscopy and imaging

Worms were anaesthetized using 20 mM levamisole, mounted on 4% agar pads and then imaged by using a Nikon Eclipse Ti microscope or Olympus FV1000a confocal microscope. L4 worms were used to record phasmid dendrite length statistics. The fluorescence intensity was measured using NIS-Elements software. The relative fluorescence intensity of cilia was calculated by subtracting the average fluorescence intensity outside the cilia from the average fluorescence intensity inside the cilia.

### Immunofluorescence of DYF-4::GFP

Worms harboring the DYF-4::GFP construct were placed on a glass slide and squashed with a coverslip. Then, the slide was frozen in liquid nitrogen, and the coverslip was removed. Then, the sample was fixed in −20˚C methanol for 20 min, followed by fixation in −20˚C acetone for 10 min, washing in PBS, blocking in 3% BSA, and sequential incubation with an anti-GFP primary antibody (mouse anti-GFP, 1:200, 11814460001, Roche) and a corresponding secondary

antibody (goat anti-mouse Alexa Fluor 488). A Nikon Eclipse Ti-E microscope was used for observation and imaging.

## GST pull-down assay

Plasmids for the expression of His-tagged DAF-6 fragments or GST-tagged DYF-4 were constructed using pET28a or pGEX-4T-1, respectively. All plasmids were transfected into *Escherichia coli* strain BL21 (DE3) to express the proteins. Protein expression was induced with 0.5 mM IPTG at 18˚C overnight, and the proteins were then purified by using Ni-Sepharose beads (GE Healthcare) or GST Sepharose beads (GE Healthcare). GST pull-down was performed as described previously [32]. Briefly, the purified His-DAF-6 ECL1 or His-DAF-6 ECL4 protein was incubated with GST-DYF-4 or GST immobilized on glutathione Sepharose beads in binding buffer (50 mM Tris-HCl, pH 7.4, 150 mM NaCl, 1% Triton X-100, 1 mM dithiothreitol, 10% glycerol, protease inhibitors) for 4 h at 4˚C. After incubation, beads were washed 5 times with the binding buffer, after which the loading buffer was added, and the sample was boiled for 10 min. The samples were then subjected to SDS-PAGE and analyzed by western blotting with a monoclonal antibody against His.

## Transmission electron microscopy

Your adult worms were immersed in 2.5% glutaraldehyde in cacodylate buffer for 24 h at 4˚C, and then postfixed in 1% osmium tetroxide in cacodylate buffer for 4 h at 4˚C. Then samples were dehydrated in a graded series of ethanol, and washed three times with pure acetone and infiltrated with a mixture of acetone and EPON 812 resin in a graded series, and finally embedded in EPON 812 resin. Ultrathin sections (~70 nm) were cut from worm heads and collected on mesh copper grids. Then samples were stained with uranyl acetate and lead citrate and imaged with a transmission electron microscope (Hitachi H-7650; Hitachi).

## Statistical analysis

Statistical differences between two samples were analyzed by Fisher's Exact test (for categorical data) or Mann-Whitney test (for continuous data). P values >0.05 were considered to indicate a nonsignificant difference. P values < 0.001 (marked as ***) were considered to indicate a significant difference.

## Supporting information

**S1 Fig. Protein structure prediction for DYF-4.** (A) Signal peptide prediction for DYF-4 by SignalP4.1. (B) Sequence alignment of the PLAC domain. *, the conserved cysteines in the PLAC domain.
(TIF)

**S2 Fig. DYF-4 is expressed in glial cells.** (A) The subcellular localization pattern of DYF-4$^{C366Y}$::GFP stained by anti-GFP. Scale bar: 5 μm. (B) The subcellular localization pattern of DYF-4 $^{\Delta(1-16)}$::GFP. Scale bar: 5 μm. (C) The localization of DYF-4::OPT-GFP in a 3 fold embryo. White arrow indicates the localization of DYF-4 in amphid glial cells. White asterisk indicates the localization of DYF-4 between the embryo and the eggshell. (D) Subcellular localization of DAF-6::GFP in young adults of WT and *dyf-4* worms.
(TIF)

**S3 Fig. The glial compartment is compromised in *dyf-4* mutants.** (A) Fluorescence micrographs of curved cilia in *dyf-4* and *daf-6* mutant worms. Curved cilia are indicated by red

arrowheads. Scale bar = 5 μm. (B) Phasmids of WT and *dyf-4* worms expressing *F16F9.3pro*::mCherry (Red, sheath glia) and OSM-6::GFP (Green, neurons). Sheath glia cells in *dyf-4* mutants could not extend as in WT. Scale bar = 10 μm. (C) WSP-1A localization at the head and tail in WT, *dyf-4(m158)* and *daf-6(e1377)* worms. Scale bars: 2 μm. (D) Quantification of GFP::WSP-1A signal length in the amphids of WT, *dyf-4(m158)* and *daf-6(e1377)* worms. (E) Quantification of GFP::WSP-1A signal length in the phasmids of WT, *dyf-4(m158)* and *daf-6 (e1377)* worms. (F) ARX-2 localization at the head and tail in WT, *dyf-4* and *daf-6(e1377)* worms. Scale bars: 2 μm. (G) Quantification of ARX-2::GFP signal length in the amphids of WT, *dyf-4(m158)* and *daf-6(e1377)* worms. (H) Quantification of ARX-2::GFP signal length in the phasmids of WT, *dyf-4(m158)* and *daf-6(e1377)* worms. All data are presented as the mean ± SEM (n ≥ 50 for each genotype). ***P < 0.001 (Mann-Whitney test).
(TIF)

**S4 Fig. Genetic interaction between glia compartment formation regulators, DYF-7 and the transition zone in dendrite extension.** (A) Quantification of phasmid dendrite length in the indicated worm lines. The defective dendrite phenotype in *dyf-7(ns117)* mutants and *nphp-1(ok500); mks-6(gk674)* double mutants was rescued by *wsp-1a(gm324)*. Each data point represents a single measurement. n represents number of dendrites analyzed. ***P≤0.001 (Mann-Whitney test).
(TIF)

**S5 Fig. The localization of IFT components in WT, *dyf-4* and *daf-6* worms.** (A) Fluorescent micrographs and quantification of the relative fluorescence intensities in the phasmid cilia of WT, *dyf-4(m158)* and *daf-6(e1377)* worms expressing various IFT markers: IFT-B components OSM-6::GFP and OSM-5::GFP, IFT-A component CHE-11::GFP and BBsome component BBS-7::GFP. Data are presented as the mean ± SEM (n ≥ 60 for each genotype). ***P≤0.001 (Mann-Whitney test). Scale bars: 5 μm.
(TIF)

**S6 Fig. Full scans of western blots for Fig 5A and 5B.**
(TIF)

**S1 Table. Worm strains used in this study.**
(DOCX)

**S2 Table. Peptide sequence information of extracellular loops in DAF-6 and CHE-14.**
(DOCX)

## Acknowledgments

We thank the Caenorhabditis Genetics Center, the Japanese National Bioresource Project, and Dr. Guangshuo Ou for worm strains.

## Author Contributions

**Conceptualization:** Qing Wei.

**Formal analysis:** Hui Hong, Huicheng Chen, Qing Wei.

**Funding acquisition:** Hui Hong, Qing Wei.

**Investigation:** Hui Hong, Huicheng Chen, Yuxia Zhang, Zhimao Wu, Yingying Zhang, Yingyi Zhang, Zeng Hu.

**Methodology:** Yuxia Zhang, Zhimao Wu, Yingying Zhang, Yingyi Zhang, Zeng Hu, Jian V. Zhang, Kun Ling.

**Project administration:** Qing Wei.

**Resources:** Jian V. Zhang, Kun Ling, Jinghua Hu.

**Supervision:** Jinghua Hu, Qing Wei.

**Validation:** Hui Hong, Huicheng Chen.

**Visualization:** Hui Hong, Huicheng Chen, Qing Wei.

**Writing – original draft:** Jinghua Hu, Qing Wei.

**Writing – review & editing:** Hui Hong, Huicheng Chen, Kun Ling, Jinghua Hu, Qing Wei.

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
