## [Decision Letter · Decision Letter 0]

22 Dec 2020

Dear Dr Wei,

Thank you very much for submitting your Research Article entitled 'DYF-4 regulates patched-related/DAF-6-mediated sensory compartment formation in C. elegans' to PLOS Genetics.

The manuscript was fully evaluated at the editorial level and by independent peer reviewers. The reviewers appreciated the attention to an important problem, but raised some substantial concerns about the current manuscript. Based on the reviews, we will not be able to accept this version of the manuscript, but we would be willing to review a much-revised version. We cannot, of course, promise publication at that time.

Should you decide to revise the manuscript for further consideration here, your revisions should address the specific points made by each reviewer. We will also require a detailed list of your responses to the review comments and a description of the changes you have made in the manuscript. Please pay particular attention to the comments from Reviewer 3, as some of these will require further experimental work to address, e.g. analysis of the glial channel, analysis of earlier developmental stages, and additional controls for the GST pulldown studies. 

If you decide to revise the manuscript for further consideration at PLOS Genetics, please aim to resubmit within the next 60 days, unless it will take extra time to address the concerns of the reviewers, in which case we would appreciate an expected resubmission date by email to plosgenetics@plos.org.

[LINK]

We are sorry that we cannot be more positive about your manuscript at this stage. Please do not hesitate to contact us if you have any concerns or questions.

Yours sincerely,

Andrew D. Chisholm

Associate Editor

PLOS Genetics

Gregory Barsh

Editor-in-Chief

PLOS Genetics

Reviewer's Responses to Questions

**Comments to the Authors:**

Reviewer #1: From a forward genetics dye filling screen, this study identifies dyf-4 as a likely extracellular regulator of sensory neuron dendrite extension and cilium formation. Although dyf-4 had been previously described in an earlier screen, it had not been cloned. The current study’s screen also identified alleles in daf-6 that phenocopy the dendrite/cilium defects of dyf-4 worms. daf-6 (Patched homologue) is a known positive regulator of dye filling and negative regulator of amphid sensory pore size. The present study reports that DYF-4 is expressed in the sensory pore glial cells, colocalising via an interdependent manner with DAF-6 at the sensory pore part of those cells; they also report that both proteins biochemically interact. The paper also shows that dyf-4 and dyf-6 interact in a suppressive manner to control dendrite extension and cilium length/IFT with wsp-1 and lit-1, which are known suppressors of the daf-6 dye filling and amphid sensory pore size defects. From these data, the authors propose that DYF-4 is an extracellular ligand for DAF-6 at the glial cell membrane, serving to regulate the glial sensory pore channel, and control dendrite extension and cilium formation.

DYF-4 is therefore a new glial regulator of C. elegans sensory pores, joining a group that includes DAF-6, WSP-1 and LIT-1. Overall, we do not know very much about how sensory pores form, and how cell-cell communication between the glial cells and ciliated sensory neurons is coordinated during this process. Thus, the topic is certainly of importance and broad interest.

Major comments:

1. The evidence for DYF-4 being an extracellular DAF-6 ligand is not very strong. Whilst the authors show that DYF-4 can bind two of the DAF-6 ECLs and demonstrate interdependent localisations, clearly more needs to be done to demonstrate a ligand-receptor relationship. Indeed, it is not clear from the data if DYF-4 is an extracellular protein, within the sensory compartment lumen, as the images do not provide enough resolution to demonstrate this. The findings of glial cell mislocalisation for DYF-4(delta 1-16) and DYF-4(C366Y) do not prove the extracellular properties of DYF-4. Also, it would be nice to show that the DAF-6/DYF-4 interaction from worm lysates. In fairness to the authors, they state in their discussion that more work needs to be performed to prove the extracellular ligand-receptor hypothesis. Therefore, they need to be more cautious about over interpretation. For example, in lines 218-219 of the results, and line 38 of abstract. I don’t expect the authors to do any more experiments for this comment, but just tone down the language in places and make it very clear what still needs to be done to prove the ligand-receptor model.

2. The paper’s major conclusion is that DYF-4 regulates sensory compartment formation, and this in turn leads to dendrite anchoring problems. Whilst the authors show mispositioning of amphid/phasmid cilia in dyf-4 L4-aged mutant, which by implication means the corresponding pores are abnormal, there is no actual data that shows dyf-4 pore lumen morphogenesis defects, as was presented previously for dyf-6 embryos/larvae by the Shaham group. Ideally, this new work would perform some TEM analysis of the dyf-4 amphid pore lumen at early ages to help prove the cause and effect relationship that is proposed. At the very least, they should include a statement in the discussion saying that TEM should be performed and results compared to dyf-6 worms.

Minor comments:

1. Some quantification of the data in S3H should be provided, as was done in Fig. 1D.

2. Fig. S4A. What exactly does ‘severely collapsed’ mean. How was this determined for the analysis shown?

3. Line 361: I think the word ‘act’ is missing.

Reviewer #2: Sensory neurons in C. elegans extend ciliated endings into the environment by protruding through tube-like glia called the sheath and socket cells. These glia contain a sensory matrix that shapes the dimensions of the glial channel while also anchoring neuronal dendrites at the glial socket. Prior studies implicated the Patched-related transmembrane protein DAF-6 in secretion or endocytosis of sensory matrix components (or some related aspect of matrix organization). Here, Hong et al identify a secreted protein, DYF-4, as an apparent partner of DAF-6.

I found this study to be very exciting! Patched-related proteins are an intriguing group of putative transporters, probably of lipids or other hydrophobic cargo. Their regulation is not well understood, though there is precedence for them acting in tandem with a soluble partner (e.g. NPC1 acts with NPC2). The relationship between DYF-4 and DAF-6 proposed here is strongly supported by shared mutant phenotypes and shared genetic interactions with suppressors, double mutant analysis, co-localization experiments, and physical binding interactions. It is a beautiful example of the power of forward genetics to find genes that act together in a biological process.

Most of my comments below have to do with improving clarity, addressing “rigor and reproducibility” issues, and adding some better context to the discussion.

Specific comments

1. Figure 1: Rather than a defect in “dendrite extension” (as the title states), this appears to be a defect in dendrite anchoring. Does the vertical line at left represent the nose tip position? If so, it appears amphid cell bodies have migrated to the appropriate location, but dendrite tips are not anchored at the nose. Similar comment applies to the phasmid defect. Actual dendrite length or the distance from nose tip to dendrite should be quantified here to capture this major aspect of the phenotype. To better understand the mutant phenotype, it would be helpful to visualize the socket and sheath glia positions and morphology (e.g. as in Low et al 2019).

2. Figure1F: what is the “phasmid dendrite defect ratio”? It is not defined in the text or legend. Is it something different from the % of severely collapsed dendrites?

3. Figure 2 and 2F: similar comments apply here as for Figure 1. Legend: “daf-6(e1377) was previously reported”. Please cite reference for this statement.

4. Figure 3: Photos are representative of how many animals imaged for these expts? What stage is being imaged? Is DYF-4 present in the developing glia of embryos (as implied in Figure 7)? Unfortunately, it is impossible to judge if DYF-4 is extracellular or in a vesicular compartment close to the apical membrane.

5. Figure 4A,B: Photos are representative of how many animals imaged for these expts? What stage is being imaged? Given the eventual loss of the channel in daf-6 mutants, it may be more informative to look at an earlier stage of channel development, when it is still intact. The word “extracellular” appears within panel A erroneously.

6. Figure 4C,D. What specific sequences are included in the ECL1 and ECL4 constructs? I do not see this information in the Methods section.

7. Figure 5: What igbd-2 allele was used here? It should be listed in legend.

8. Figures 5, S4: How were “severely collapsed dendrites” defined?

9. Figure 7 (model): The model should more clearly show what is being newly proposed based on the current work. Currently, this model is mostly summarizing data from prior Shaham or Heiman lab papers about daf-6 or dyf-7 (which should therefore be cited in the legend). There are no data shown in this paper to address whether the glial channel is initially open or whether socket and sheath glia detach in dyf-4 mutants. The placement of DYF-4 inside the sheath cell is particularly confusing, and it should be more clearly explained – are you proposing that DYF-4 is in a vesicular compartment (e.g. like NPC2) or in the extracellular matrix? Perhaps both models should be shown.

10. It would be informative to mention NPC1/NPC2 precedent in the Discussion.

11. Table S1: What is IsOSM-6? This is non-standard nomenclature. Also, Ex transgene names should have lab and numerical identifiers.

12. Statistical methods should be re-evaluated in consultation with a statistician. The authors use a student’s t-test for both categorical and continuous data. Categorical data (e.g. dyf or not dyf; severely collapsed or not) should be analyzed with a Fisher’s Exact test instead. A t-test is appropriate for continuous data (e.g. distance measurements), but only if the data have a normal distribution and similar variance between groups, which doesn’t always seem to be the case here. Consider a non-parametric test such as Mann-Whitney instead.

13. There are many typos or grammatical errors, some of which are listed below. A copy editor will be needed.

line 50: “how DAF-6 functions and be regulated”

lines 99-100: glial channel formation requires the dendrite tip anchoring protein DYF-7 (Low et al., 2019), sheath cells fail to undergo extension in dyf-7 mutants (Heiman and Shaham, 2009).

Line 150: “defects described here is a new phenotype”

Line 180: “DYF-4 is at least partially colocalizes with DAF-6”

Line 188: “DYF-4 function in a same pathway as DAF-6”

Line 203: “is the results of”

Lines 248-249: “results in severely dendrite elongation defects …”

Reviewer #3: This manuscript presents the identification of dyf-4, a gene involved in sensory dendrite/cilia development. It encodes a secreted protein that the authors show is expressed in glia and localizes to a glial channel around the dendrites/cilia. The authors identify an interesting relationship with DAF-6, a glial multipass transmembrane protein that has been previously shown to shape the glial channel. Specifically, each protein fails to localize normally in the absence of the other, and the authors use a GST pull-down assay to show a physical interaction between DYF-4 and the extracellular loops of DAF-6. The identity of DYF-4 and its potential interaction with DAF-6 are interesting, however the data seem preliminary and do not support the main conclusions of the paper.

1. The authors interpret their results in terms of morphogenesis of the glial channel. However, the data mostly relate to dendrite length and the glial channel is not examined directly at all. This is a major shortcoming. The authors seem to use dendrite length as a proxy for glial channel morphogenesis but it is not clear what the glial channel defects are, when they appear, or how (if) they relate to dendrite/cilia development.

2. The authors interpret their results in terms of developmental defects. However, the data relate to L4 stage phenotypes. Embryos (or even L1s) are not examined. This is another major shortcoming, especially in light of the relationship to DAF-6, which has progressively more severe defects, becoming extremely disorgnized by L4/adult. daf-6 dendrites are short only in adults, not in L2 larvae (Albert et al. 1981). It is not clear if dyf-4 affects dendrite/cilia development directly or if these are secondary defects that develop later as the structure becomes increasingly disorganized, as is the case for daf-6.

3. Many of the results are inconsistent with previously published work. In principle this is not an issue but there should be some attempt to acknowledge/reconcile the differences. The defining phenotype of daf-6 mutants is the accumulation of vacuoles/enlarged secretory structures at the distal glial channel which does not seem to be seen here. Conversely the authors report shortened cilia which is not consistent with previous studies. Can the authors account for these differences?

4. The localization defects are interesting but are more consistent with a chaperone role in which DYF-4 and DAF-6 mutually require each other for trafficking to the cell surface. However the authors' interpretation is that DYF-4 is an extracellular ligand for DAF-6, acting in the glial channel, despite the fact that DYF-4 cannot rescue when expressed from the neurons. This is very confusing and seems to contradict their results.

5. The genetic interaction experiments are also very confusing. dyf-4 and daf-6 do not interact genetically with dyf-7, but suppressors of dyf-4 and daf-6 suppress dyf-7. What is the significance of this? The authors conclude "there are genetic interaction between glia compartment morphogenesis, dendrite anchoring complex and the TZ in dendrite extension". But, it is not clear that any of these mutants are affecting the same steps – for example, dyf-7 dendrite extension defects occur in embryos, and daf-6 dendrite length defects do not occur until late larval development. These experiments also use a hypomorphic allele of dyf-7 which complicates the analysis. All of these suppressor/enhancer experiments seem quite preliminary and inconclusive.

6. The GST pull-down experiments are potentially promising but also seem preliminary and to lack appropriate controls. It is curious that both the extracellular loops of DAF-6 independently bind DYF-4 equally well. Perhaps DYF-4 is just "sticky"? It would help to include some negative controls, for example extracellular loops of related proteins like CHE-14 that are not expected to bind DYF-4, or point mutations in the DAF-6 loops that have been shown to disrupt its function. S253P in loop 1 and F568L in loop 4, which disrupt localization of DAF-6 similarly to loss of dyf-4 (Perens et al. 2005), would be especially good candidates to investigate.

7. The model figure seems to place WSP-1 and LIT-1 upstream of DAF-6 and DYF-4. This is not consistent with the genetic data, in which loss of these WSP-1/LIT-1 suppresses loss of DAF-6/DYF-4. The model figure itself adds very little to the paper other than re-stating the phenotypes, while omitting the key features of the daf-6 mutant (over-expansion of the sheath pore in the embryo).

**Have all data underlying the figures and results presented in the manuscript been provided?**

Reviewer #1: Yes

Reviewer #2: **No: **numerical data underlying graphs are not provided

Reviewer #3: Yes

PLOS authors have the option to publish the peer review history of their article (what does this mean?). If published, this will include your full peer review and any attached files.

Reviewer #1: No

Reviewer #2: No

Reviewer #3: No

---

## [Decision Letter · Decision Letter 1]

15 Apr 2021

Dear Dr Wei,

Thank you very much for submitting your Research Article entitled 'DYF-4 regulates patched-related/DAF-6-mediated sensory compartment formation in C. elegans' to PLOS Genetics.

The manuscript was fully evaluated at the editorial level and by independent peer reviewers. The reviewers appreciated the attention to an important topic but identified some concerns that we ask you address in a revised manuscript

We therefore ask you to modify the manuscript according to the review recommendations. Your revisions should address the specific points made by each reviewer.  We agree with the reviewers' comments that data shown in the rebuttal letter needs to be incorporated into the final manuscript, most likely as additional supplemental data.

The C. elegans transgenic nomenclature needs to conform with standards in this field, specifically all transgenes should have unique identifying Ex or Is numbers. The Methods or Supplementary Table should also state the concentrations of experimental and coinjection marker DNAs used in transgene generation.

[LINK]

Yours sincerely,

Andrew D. Chisholm

Associate Editor

PLOS Genetics

Gregory Barsh

Editor-in-Chief

PLOS Genetics

Reviewer's Responses to Questions

**Comments to the Authors:**

Reviewer #1: Authors have done a good job addressing my comments.

Reviewer #2: Hong et al provide compelling evidence that the secreted protein DYF-4 functions with the Patched-related protein DAF-6 to affect the morphology of C. elegans glial sensory channels and their cognate sensory neurons. The discovery of a DAF-6 partner is very significant given the widespread importance of Patched, Dispatched, NPC1, and other Patched-related proteins; therefore this paper should be of broad interest.

The revision now provides more direct evidence for glial channel defects by visualizing the sheath and socket glia with fluorescent markers and by TEM; this confirms that the dyf-4 and daf-6 phenotypes are really similar.

The authors have also removed claims for a ligand-receptor relationship between DYF-4 and DAF-6, and instead more appropriately consider several possible models for their relationship.

There are remaining issues:

There are many places where the authors provided clarifying information in the "response to reviewers", but then failed to include that information in the actual manuscript. This information is needed for readers to understand (and believe) results shown in the paper.

1. Figure 1B, 2B legends should explain that the vertical line at left represents the nose tip position.

2. Figure1F, 5C, 6D: The “phasmid dendrite defect ratio” and/or "severely collapsed dendrites" should be defined in the Methods or legends.

3. Regarding the timing of the phenotype, L1 and L2 data shown in the "response to reviewers" (Rev 3 Question 2) should be shown and discussed in the manuscript (could be supplemental).

4. The DYF-4::OPT-GFP embryo image included in the "response to reviewers" should be shown in Figure 3 or S2, as it clearly shows that the protein is present during embryonic development and is secreted, accumulating between the embryo and the eggshell.

5. Figures 3 and 4: The legends should list the stage and say that these are "representative of more than 100 worms" (or whatever is the case).

6. Figure 3E. The specific sequences included in the ECL1 and ECL4 constructs should be listed in the Methods section or in a supplemental Figure/Table.

7. Figure 3E. The control pull-down experiments done with CHE-14 (shown in response to reviewer 3) should also be shown in the manuscript (could be supplemental).

8. Table S1: The OSM-6::GFP nomenclature was fixed, but a large number of QW strains still use non-standard nomenclature. All Ex transgenes should be named with lab and numerical identifiers.

9. There are still typos or grammatical errors, some of which are listed below.

Fig. S6: The figure panel itself says it refers to Fig. 4C, whereas the legend refers to Fig. 3E.

Lines 146-148: "It is likely that the abnormal morphology of glial channels changes the anchoring position of the tips of dendrites, which indirectly leads to the shortening of dendrites". This sentence sounds like pure speculation - please add "see below" or give some indication that data about glial channel morphology will be shown later in the manuscript.

Line 190: "only partial of amphids"

Line 202: "Dendrite defects are much more severed" (severe)

10. Some of the data in the Supplemental Figures are quite important and really should be shown in a main Figure instead.

For example, rescue data in Fig. S2B (text lines 156-159) argue for a glia cell-autonomous requirement for DYF-4, which is more consistent with a role in DAF-6 trafficking or some other vesicular process, and less consistent with a ligand-receptor relationship on the external cell surface (as pointed out by Reviewer 3). The data could be highlighted and discussed in this context.

DYF-4::OPT-GFP imaging data in Fig. S2F shows the punctate localization that leads authors to propose DYF-4 might be in vesicles (text lines 177-178).

Double mutant data in Fig. S3I is critical for the conclusion that dyf-4 and daf-6 act in the same pathway.

Reviewer #3: The authors have clearly made a diligent attempt to respond to the comments. The manuscript has been improved by their revisions. By being somewhat more cautious in its conclusions and providing better support for the central claims, the revised manuscript really highlights the strengths of this exciting story.

Regarding my previous major comments:

1. The EM and other additional data related to visualizing defects in the glial channel have improved the story.

2. Regarding timing of the phenotypes, the data that are shared privately in the reviewer response are excellent. In my view this result, showing that dendrite defects only appear at the L2 stage, is essential to interpreting the results. **It needs to be included in the manuscript.**

3. These clarifications and additions are very helpful.

4. The revised discussion of the DYF-4-DAF-6 relationship is a major improvement.

5. The discussion of genetic interactions remains somewhat confusing, and possibly misleading. It is really important to clarify in the text that dendrite defects in dyf-7, TZ mutants, and daf-6/dyf-4 arise at different developmental stages, suggesting that any genetic interactions among them would be indirect.

The authors respond to my comments, "Although DYF-7 anchoring complex forms in the embryo, glia channel defects in later stage will definitely change the position or composition of extracellular matrix which will in turn affect the position of dendrites." My understanding is that DYF-7 controls dendrite length by preventing rupture of the developing sheath-socket-hyp epithelium in the embryo. It is not clear to me how the ECM would affect dendrite anchoring after the epithelium has already ruptured. It seems like these genes are acting at different developmental stages, and probably through different mechanisms.

6. The additional figure shared privately in the rebuttal letter is very helpful and strengthens an important conclusion of the paper. Because it addresses an important point that many readers will consider, I would **strongly encourage** the authors to add it to the manuscript.

I have a few other minor comments that might improve the manuscript, but these should be handled at the authors' discretion and are not an impediment to publication:

1. In the introduction, perhaps revise to "the distal portions of *most* cilia extend through a pore formed by the membrane of the socket cell" to clarify that this is not the case for some cilia (AWA, AWB, AWC, AFD).

2. Several minor typos – "compartmentation" for "compartmentalization", "severed" for "severe", "socked" for "socket", "Gila" for "Glia" in Fig 7.

3. It is surprising that DAF-6 ECL1 and ECL4 both bind DYF-4. It would help to note whether these sequences share any motifs (does the 12 TM structure of DAF-6 look like a duplication of a more ancestral 6 TM sequence, with ECL4 homologous to ECL1?). As noted, the additional CHE-14 control included in the rebuttal letter would really help to shore up the specificity of this result.

4. "the localization of DYF-4::OPT-GFP is basically normal, only partial of amphids had abnormal DYF-4 signals". I think this should read "only a fraction of amphids", and it would be very helpful to cite what proportion (even a ballpark estimate, e.g. "~5%", "<20%", etc.).

5. In Fig. 4B, a zoomed-out view would help – is DAF-6 mostly stuck in the cell body? I am confused why a 12-TM protein is giving a diffuse cytoplasmic signal. It looks like soluble GFP. Perhaps clarify in the text what we are looking at.

6. In Fig. 5, the difference between % Dye-filling and % severely collapsed is potentially confusing. For example, wsp-1a; dyf-4 appears to be almost fully rescued for dendrite collapse but nevertheless fails to dye-fill. I think this is because many dendrites are of intermediate length – it might help to clarify this in the text. On a related note, it would help to be consistent between Fig. 5A and 5C in whether you show the fraction that are normal or defective (so WT would be 100% or 0% in both plots).

7. The claim that "the actin cytoskeleton is likely the major downstream functional site of DYF-4-DAF-6-regulated glial function" seems like it goes far beyond what the data support. There does not seem to be any evidence yet to distinguish whether WSP-1 acts downstream of DYF-4/DAF-6 rather than in a parallel pathway. For example, if dyf-4/daf-6 defects arise by excess vesicle fusion to the forming channel lumen, then wsp-1 mutants might suppress this defect by reducing vesicle delivery to the apical surface. This seems quite possible, and dyf-4/daf-6 would not affect actin in this case at all. I would be very careful here.

8. In the model drawn in Fig. 7, showing DAF-6 as an inhibitor of glia channel morphogenesis suggests that loss of DAF-6 should improve channel morphogenesis – but in fact, loss of DAF-6 disrupts morphogenesis. I understand what you are trying to convey, but something like "glial channel growth" or "expansion" might be better.

**Have all data underlying the figures and results presented in the manuscript been provided?**

Reviewer #1: Yes

Reviewer #2: **No: **numerical data underlying graphs has not been provided

Reviewer #3: Yes

PLOS authors have the option to publish the peer review history of their article (what does this mean?). If published, this will include your full peer review and any attached files.

Reviewer #1: **Yes: **Oliver Blacque

Reviewer #2: No

Reviewer #3: No

---

## [Editor Report · Decision Letter 2]

24 May 2021

Dear Dr Wei,

We are pleased to inform you that your manuscript entitled "DYF-4 regulates patched-related/DAF-6-mediated sensory compartment formation in C. elegans" has been editorially accepted for publication in PLOS Genetics. Congratulations!

Yours sincerely,

Andrew D. Chisholm

Associate Editor

PLOS Genetics

Gregory Barsh

Editor-in-Chief

PLOS Genetics

Comments from the reviewers (if applicable):

**Data Deposition**

http://datadryad.org/submit?journalID=pgenetics&manu=PGENETICS-D-20-01746R2

**Press Queries**

---

## [Editor Report · Acceptance letter]

8 Jun 2021

PGENETICS-D-20-01746R2 

DYF-4 regulates patched-related/DAF-6-mediated sensory compartment formation in *C. elegans*

Dear Dr Wei, 

We are pleased to inform you that your manuscript entitled "DYF-4 regulates patched-related/DAF-6-mediated sensory compartment formation in *C. elegans*" has been formally accepted for publication in PLOS Genetics! Your manuscript is now with our production department and you will be notified of the publication date in due course.

With kind regards,

Zita Barta

PLOS Genetics

On behalf of:
